# In Situ IR Study on Effect of Alkyl Chain Length between Amines on Its Stability against Acidic Gases

**Rose Mardie Pacia, Clinton Manianglung and Young Soo Ko *** 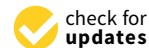

Department of Chemical Engineering, Kongju National University, 275 Budae-dong, Seobuk-gu,
Cheonan-si 31080, Korea; rppacia@smail.kongju.ac.kr (R.M.P.); clintonmanianglung@smail.kongju.ac.kr (C.M.)
* Correspondence: ysko@kongju.ac.kr; Tel.: +82-41-521-9364

**Abstract:** For the $CO_2$ capture process via the cyclic adsorption/desorption method, one emerging catalyst is the use of amine-functionalized silica. This study focused on comparing the $CO_2$ capture performance of diamines with ethyl and propyl spacers and the degradation species formed after long-term exposure to various acidic gases such as $SO_2$ and $NO_2$ at elevated temperatures. Adsorbents were prepared via the incipient wetness technique and then subjected to thermogravimetric measurements and in situ FT-IR analyses. 2NS-P/Kona95, which contains a propyl spacer, showed fewer degradation species formed based on its IR spectra and better stability with its long-term exposure to various acidic gases. Thus, the incorporation of amines with a large number of nitrogen groups of propyl or longer spacer length could be a promising $CO_2$ capture material.

**Keywords:** $CO_2$ capture; catalytic amines; diaminosilane; propyl spacer; acidic gas; degradation; in situ IR

## 1. Introduction

A significant amount of research has been conducted on ways to mitigate the continuous increase in $CO_2$ emissions [1]. Currently, there are three major technologies that are considered to be viable processes to address $CO_2$ emission issues, one of which is postcombustion capture (PCC), which is a set of end-pipe technologies that appears to be a promising method of capturing $CO_2$ from combustion process exhaust [2,3]. Under PCC, various techniques such as membrane separation, cryogenic distillation, absorption using aqueous solutions, and catalytic adsorption using organic/inorganic hybrid adsorbents have been developed to achieve an optimal and efficient $CO_2$ capture process [4]. Though the absorption process using aqueous amines as a solvent is the most mature method, it has several drawbacks, such as the high amount of energy required for regeneration, low $CO_2$ loading capacity, high equipment corrosion rate, large equipment size requirements, and loss of solvent in regeneration [5–8]. Recently, adsorption via organic/inorganic hybrid adsorbents has received widespread attention for its ability to reversibly capture $CO_2$ from flue gas streams and its advantages of reduced energy requirements for regeneration, greater capacity, higher selectivity, and ease of handling [9].

Several materials have been investigated as supports for preparing these organic/inorganic hybrid adsorbents, such as activated carbon, carbon nanotubes, zeolites, molecular sieves, metal–organic frameworks (MOFs), alkali metal carbonate-based solid sorbents, and amine-functionalized silica [10–12]. Among these, the latter has emerged as a promising material, as it possesses a high adsorption capacity, fast adsorption kinetics, facile regeneration, and high selectivity [12,13]. This type of adsorbent is prepared by introducing an organic amine site to inorganic silica supports with a porous structure that can expose the basic amine groups as adsorption sites of acidic $CO_2$. One of the most widely used amines is polyethyleneimine (PEI) and diaminosilane, which contains diamine

units with ethyl spacers, due to its high-quantity amine groups that act as adsorption sites for catalysis. Aminosilanes have been employed to functionalize the surface of the catalyst support to attach transition metal catalysts for a variety of catalytic reactions such as polymerization, hydrogenation, and so forth.

In the desorption process, temperature-swing adsorption (TSA) is an attractive approach for PCC using adsorbents [14] because of its reduced cost due to the low-temperature capture step. In this process, adsorption takes place at a low temperature and desorption is carried out by increasing the temperature [15]. It usually utilizes pure and dry $CO_2$ as a sweep gas in the desorption stage [16,17]. This process yields a highly pure $CO_2$ product stream, but complete desorption is not achieved because the driving force for desorption is not sufficient and irreversible species with various gases are generated, causing severe long-term instability during cyclic operations. Moreover, since TSA requires the continuous treatment of the adsorbent at an elevated temperature and with the presence of $CO_2$, studies on the behavior of the adsorbent in these conditions have been conducted. Typical flue gas composition is 10%–15% $CO_2$, 3%–10% $O_2$, 4%–5% $H_2O$, 2000 ppm $SO_2$ (without desulfurization), 1500 ppm NO (without catalytic reduction), and the balance is $N_2$ [18]. In order to apply the amine-containing adsorbents industrially, performance under realistic conditions must be assessed. Therefore, the stability of the adsorbents in these gases must be evaluated. Studies have proved that in the case of primary amines, urea species were generated, and with $O_2$, in the case of the secondary amines, amide groups were formed at high temperatures [17]. Moreover, extensive research has been conducted to determine the effect of increasing the number of amine groups with the capture process, but only a few works have reported the effect of amine groups with ethyl and propyl spacers [19–21]. According to Pang et al., poly(propylene imine) (PPI)-silica-supported adsorbents showed increased tolerance during the regeneration step and the presence of secondary amines did not necessarily yield to degradation species at elevated temperatures [22]. Furthermore, performance of diamines with propyl spacers on oxidative and acidic gases has not been fully investigated [23]. It is quite important to study and improve the stability of amine-functionalized silica adsorbents against acidic gases and $CO_2$ for both industrial and academic purposes.

In this study, we aimed to further investigate using in situ IR the effect of diamine-functionalized silica with propyl spacers (N-(3-(trimethoxysilyl)propyl)propane-1,3-diamine, as 2NS-P) in comparison with those with ethyl spacers (N-[3-(trimethoxysilyl)propyl]ethylenediamine, as 2NS) at various acidic gas conditions. The adsorbents were prepared by the incipient wetness technique to functionalize diaminosilane on silica (Kona95), which were named 2NS-P/Kona95 and 2NS/Kona95. The $CO_2$ adsorption performance of the adsorbents was analyzed using thermogravimetric analysis. Moreover, using in situ FT-IR analysis, the degradation species generated after the long-term exposure of the adsorbents to $CO_2$, $O_2$, $SO_2$, and $NO_2$ gases were also determined.

## 2. Results and Discussion

### 2.1. Characterization of the Adsorbents

The physical characteristics of the bare silica support and amine-functionalized silica adsorbents are summarized in Table 1, along with the silane content of the materials. Physical properties of the support and adsorbents were analyzed by the Brunauer–Emmett–Teller (BET) method. The incipient wetness technique, which was the method employed in the incorporation of aminosilanes in silica, utilizes water present inside the pores of silica for the substitution of the alkoxy group of aminosilane to hydroxyl and additional polycondensation reactions among OH-substituted aminosilanes [24,25]. Figure 1 presents a representation of the functionalization inside the pores of the silica support. Therefore, the reduction of the surface area and pore volume of the bare silica support marked the successful incorporation of aminosilanes to silica.

**Table 1.** Textural properties of the support and amine-functionalized silica from $N_2$ sorption isotherms at −196 °C.

| Sample | Silane Content (mmol/g) | $S_{BET}$ (m$^2$/g) [a] | $V_p$ (cm$^3$/g) [b] |
|---|---|---|---|
| Kona95 | - | 253 | 0.765 |
| 2NS/Kona95 | 3.19 | 18 | 0.105 |
| 2NS-P/Kona95 | 2.50 | 23 | 0.114 |

[a] Surface area determined by the BET method at $P/P_o$ = 0.05–0.20. [b] Total pore volume determined as the amount of $N_2$ adsorbed at $P/P_o$ = 0.99. Abbreviations: 2NS— N-[3-(trimethoxysilyl)propyl]ethylenediamine; 2NS-P— N-(3-(trimethoxysilyl)propyl)propane-1,3-diamine.

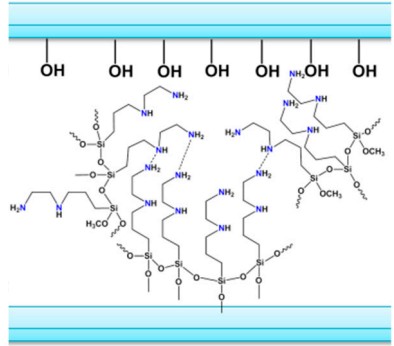

**Figure 1.** Amine functionalization inside the silica pores by the incipient wetness technique.

## 2.2. Thermogravimetric Analysis

### 2.2.1. Degradation to $CO_2$

Much attention has been paid to the effect of $CO_2$ on adsorbents primarily because of its relatively large proportion in flue gas compared with other acidic gases. In order to measure the $CO_2$ sorption capacity during and after $CO_2$ exposure, TGA was carried out. Table 2 summarizes the percent (%) decrease in $CO_2$ sorption capacity of the adsorbents upon exposure to 100% $CO_2$ for 24 h at different temperatures. Since cyclic urea is known to form from the reaction of $CO_2$ with amines at elevated temperatures, 110, 130, and 150 °C were chosen.

**Table 2.** Percent decrease in the capacity of the adsorbents exposed to 100% $CO_2$ at 110, 130, and 150 °C for 24 h in TGA.

| Adsorbent | Silane Content (mmol/g) | Fresh $CO_2$ Sorp. (wt %) | Exposure at 110 °C | | Exposure at 130 °C | | Exposure at 150 °C | |
|---|---|---|---|---|---|---|---|---|
| | | | $CO_2$ Sorp. (wt %) | % Decrease | $CO_2$ Sorp. (wt %) | % Decrease | $CO_2$ sorp. (wt %) | % Decrease |
| 2NS/Kona95 | 3.19 | 6.70 | 1.27 | 81 | 0.99 | 85 | 0.87 | 87 |
| 2NS-P/Kona95 | 2.50 | 6.70 | 2.17 | 68 | 1.89 | 72 | 1.74 | 74 |

It is noticeable that huge amount of $CO_2$ uptake was lost at all temperatures for both adsorbents. Additionally, the percent decrease in $CO_2$ capacity increased as a function of temperature. For further analysis, percent mass change was plotted against time during the $CO_2$ exposure of the adsorbents at all temperatures (Figure 2). The sudden rise in mass was an effect of the sudden exposure to an abundant amount of adsorbable gas ($CO_2$). During the exposure, the net amount of $CO_2$ adsorbed at 110, 130, and 150 °C was 1.26, 1.76, and 1.9 wt %, respectively, for 2NS/Kona95, while 1.69, 2.26, and 2.20 wt % were recorded for 2NS-P/Kona95. The amount of $CO_2$ adsorbed was found to increase as a function of temperature. This was an effect of the increased diffusion of $CO_2$ molecules into the adsorption sites and amine stretching, which enhanced the catalytic adsorption activity. Organic amine stretching occurs at a high temperature, which can cause an increase in the number of exposed

adsorption catalytic sites [26]. However, at 130 and 150 °C, only small differences were observed for both of the adsorbents. For all temperatures, 2NS-P/Kona95 demonstrated a higher $CO_2$ uptake than 2NS/Kona95. This characteristic could be attributed to its higher surface area than 2NS/Kona95, which could be correlated with the laid out adsorption sites available for $CO_2$ capture. This could be explained in parallel with the results of linear and dendritic aminopolymers with ethyl and propyl spacers [22]. According to their results, aminopolymers with a longer alkyl spacer length had higher amine efficiency ($CO_2$/N) due to the increased catalytic adsorption activity and basicity ($pK_b$) and decreased neighbor interactions; consequently, they demonstrated increased $CO_2$ capture capacity and catalytic efficiency [22,27,28].

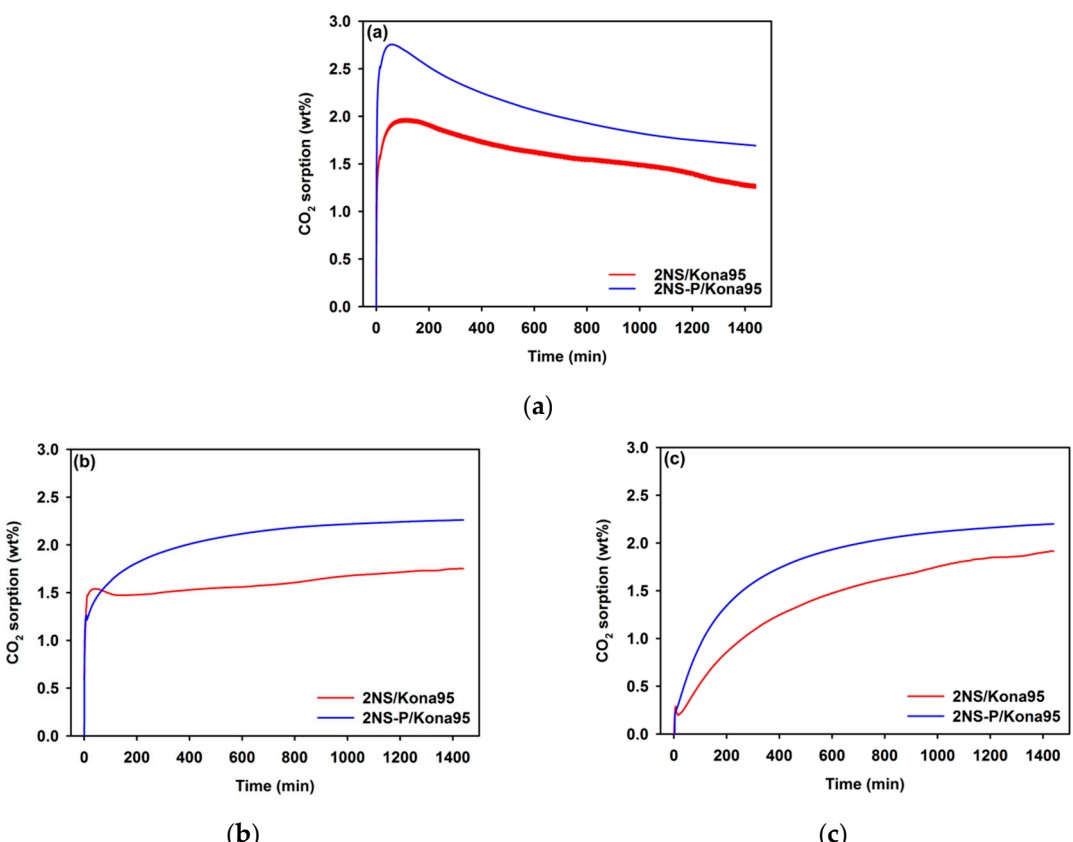

**Figure 2.** Change in mass of the adsorbents upon exposure to 100% $CO_2$ at (**a**) 110, (**b**) 130, and (**c**) 150 °C.

### 2.2.2. Degradation to $O_2$ and Acidic Gases

Aside from $CO_2$, a considerable amount of $O_2$ and a minimal content of acidic gases ($SO_x$ and $NO_x$) are also present in flue gases. These gases react with the amines of the adsorbents and can therefore affect the performance of the adsorbent. Commonly, concentrated acidic gases act as a poison to the catalyst, where they decrease the reaction rate of the active sites. In this subsection, long-term exposure to air (21% $O_2$), $SO_2$, and $NO_2$ (50 and 200 ppm) is discussed in terms of the changes in the adsorbents' $CO_2$ sorption capacity. The stability of the adsorbents against oxidative degradation was assessed in a thermogravimetric analyzer by flowing air (21% $O_2$/$N_2$) into the system at 110 °C for 24 h. The results of this analysis are presented in Figure 3.

The $CO_2$ sorption capacity of 2NS/Kona95 decreased 49%, mainly due to the oxidation of its secondary amines to form amide. This was supported by the FT-IR results, which showed the appearance of a peak at 1680–1670 $cm^{-1}$, attributed to the stretching mode of C=O of the amide [29–31]. 2NS-P/Kona95, also having secondary amines, was expected to experience the same oxidative degradation. However, this was observed to occur to a much lesser extent (16% loss) compared with 2NS/Kona95. This is consistent with the result of Pang et al., where the adsorbent with ethyl

spacers (triethylenetetramine) lost 90% of its capture capacity, while the adsorbent with propyl spacers (tripropylenetetramine) lost only 20% of its capture capacity [22]. In another study [32], where amine solutions of ethylenediamine (EDA) and 1,3-diaminopropane (1,3-DAP) were analyzed, EDA was found to be more susceptible to oxidative degradation, showing an 80% loss of the initial amount of amine compared with 1,3-DAP, which had negligible degradation.

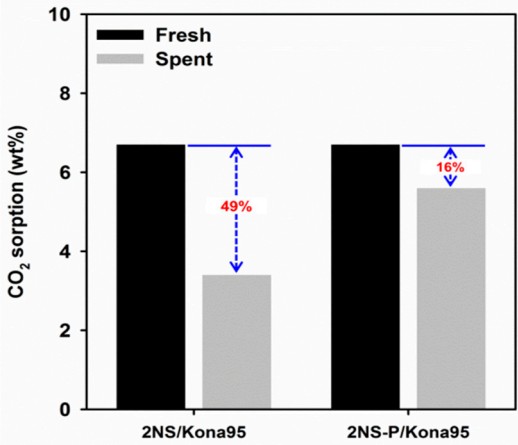

**Figure 3.** $CO_2$ sorption capacity of Kona95-supported adsorbents before and after exposure to 21% $O_2$ for 24 h at 110 °C.

Since minimal amounts of acidic gases are still present in flue gas, even though further purification systems are installed in plants, we also needed to investigate the detrimental effect of acidic gases ($SO_2$ and $NO_2$) on the adsorbents. The $CO_2$ sorption capacities of the spent adsorbents were compared with the fresh ones and the results for $SO_2$-degraded adsorbents are summarized in Figure 4. As expected, higher degradation was experienced at a higher $SO_2$ concentration of 200 ppm. The same behavior was observed for $NO_2$ gas treatment. Moreover, the adsorbent with a longer diaminosilane length (2NS-P/Kona95) was found to be more stable than the shorter one. The preferential catalytic oxidation of secondary amines by acid gases to form nitro groups paved the way for the degradation of 2NS/Kona95 due to its shorter chain length, which could provide more accessible contact for $NO_2$ and amine.

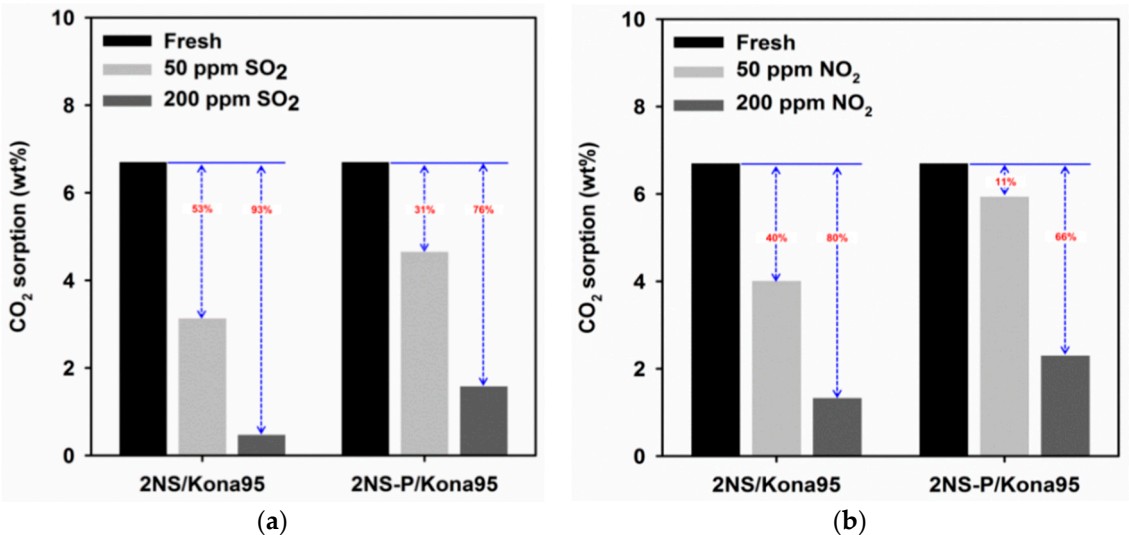

**Figure 4.** $CO_2$ sorption capacity of Kona95-supported adsorbents before and after exposure to 50 and 200 ppm of (**a**) $SO_2$ and (**b**) $NO_2$ at 110 °C.

### 2.3. *In Situ FT-IR Spectra of the Adsorbents*

#### 2.3.1. Degradation to $CO_2$

To further analyze what degradation species may have been formed on the adsorbents, in situ FT-IR analysis was conducted by scanning the spectra hourly after long-term exposure to various gases. The peak assignments for the degradation species induced by $CO_2$ are listed in Table 3. The changes in the structure of the adsorbent during the 24 h of $CO_2$ exposure are displayed in Figure 5. After $CO_2$ treatment, peaks at 1700 and 1495 cm$^{-1}$ appeared in 2NS/Kona95. These peaks correspond to the C=O and C–N stretches of cyclic urea. At higher temperatures, the dominating species for 2NS/Kona95 was cyclic urea. For 2NS-P/Kona95, peaks at 1665, 1635, 1520, and 1350 cm$^{-1}$ appeared, corresponding to the vibrations of carbamic acid and carbamate. Moreover, cyclic urea peaks were not observed. Certainly, diamine with ethyl spacers was more susceptible to cyclic urea degradation than diamine with propyl spacers.

**Table 3.** IR peak assignments for the species formed from the reaction between $CO_2$ and amine.

| Wavenumber (cm$^{-1}$) | Assignment | Species |
|---|---|---|
| 1705–1700 | C=O stretch | Cyclic urea |
| 1700–1680 | C=O stretch | Carbamic acid |
| 1635–1625 | $NH_3^+$ asym. def | $NH_3^+$ |
| 1601–1590 | NH deformation | $NH_2$ |
| 1570–1545 | $COO^-$ asym. stretch | Carbamate |
| 1565–1560 | C–N stretch | Cyclic urea |
| 1520–1515 | NH deformation C–N stretch | Silylpropyl carbamate |
| 1510 | NH def./ C–N stretch | Carbamate |
| 1550–1485 | $NH_3^+$ sym. def. | $NH_3^+$ |
| 1500–1495 | C–N stretch | Cyclic urea |
| 1490–1480 | $NCOO^-$ vibration | Carbamate |
| 1440–1410 | $COO^-$ sym. stretch | Carbamate |
| 1380 | $COO^-$ sym. stretch | Carbamate |
| 1320 | $NCOO^-$ vibration | Carbamate |

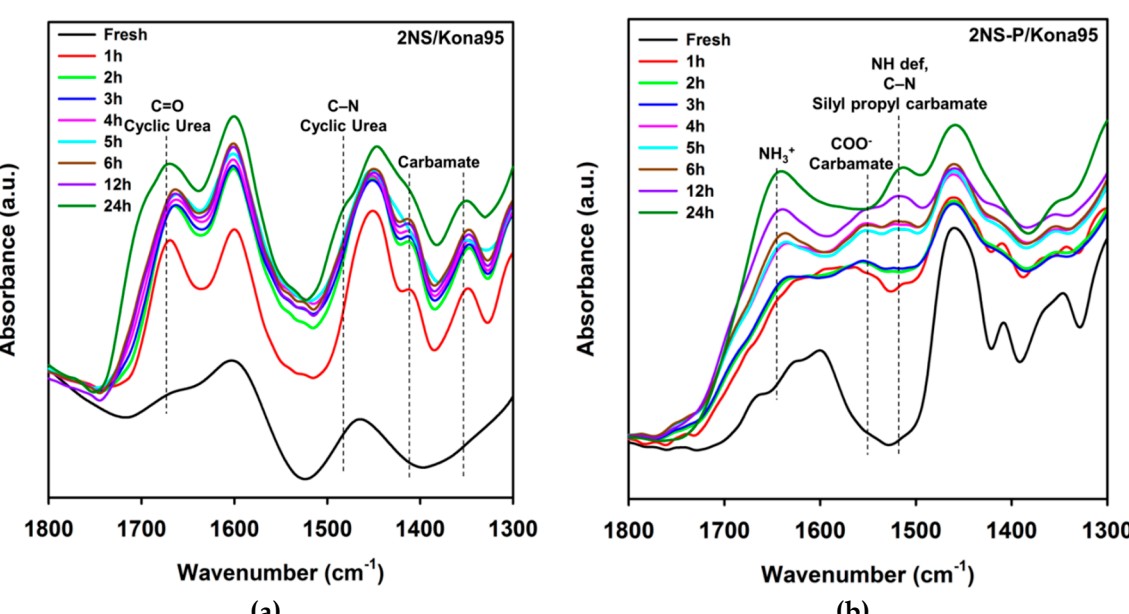

**Figure 5.** Changes in the in situ FT-IR spectra of (**a**) 2NS/Kona95 and (**b**) 2NS-P/Kona95 exposed to 100% $CO_2$ at 110 °C.

In order to provide a comparison with the work of Pang et al. [22], 110 °C was chosen for further evaluation. There was a drastic change observed in the spectrum of 2NS/Kona95 during the first hour. The growth of the peak corresponding to the formation of cyclic urea was evident and accompanied by a change in the color of the adsorbent from white to yellow. This means that the sudden and continuous increase in mass during the first hour was due to the formation of cyclic urea. Carbamate species were also observed, as evidenced from the peaks at 1410 and 1350 $cm^{-1}$. As the exposure progressed, minimal changes in the intensities of cyclic urea were seen, with minimal lowering of carbamate peaks. Therefore, the lowering of the $CO_2$ sorption capacity of 2NS/Kona95 was due to the irreversible formation of cyclic urea. 2NS-P/Kona95 also experienced a drastic change in its spectrum for the first hour of exposure. This was related to the formation of $NH_3^+$ species of the adjacent amine when the other amine formed carbamate, accompanied by the lowering of N–H peaks at 1600 $cm^{-1}$. Moreover, the already bonded amines with $CO_2$ in the form of carbamates (1545 and 1515 $cm^{-1}$) paved way for decreasing $CO_2$ sorption after exposure.

In a $CO_2$-induced deactivation study [19] of liquid EDA and 1,3-DAP, EDA and 1,3-DAP were proposed to undergo carbamate formation followed by intermolecular cyclization, accompanied by dehydration, to form cyclic urea. The pathway could be the reaction of carbamate with another amine to form linear urea, which subsequently reacts with $CO_2$ to form a large molecule of urea. However, it is also known that a ring closure does not occur significantly in amine with a 1,3-DAP unit [33,34]. The formation of cyclic urea of supported amine adsorbents with an EDA unit proceeds in a similar way and produces a similar product (five-membered ring of urea). Consequently, the mechanism of supported amine adsorbents with a 1,3-DAP unit can also be associated with the liquid 1,3-DAP mechanism. Since carbamate peaks, instead of cyclic urea peaks, were observed in 2NS-P/Kona95 (containing 1,3-DAP unit), it could be inferred that under the conditions in this study, intermolecular cyclization and dehydration did not proceed in order to form the corresponding cyclic urea.

### 2.3.2. Degradation to $O_2$ and Acidic Gases

To further analyze the behavior of the adsorbents, in situ FT-IR measurements were also done for the oxidative and acidic gas treatments of the adsorbents. Figure 6 shows the IR spectra obtained for the oxidative treatment, revealing a growth of the band corresponding to C=O of amide at 1670 $cm^{-1}$. After the first hour of exposure, amide was already formed for both of the adsorbents, but the rate of formation of amide for 2NS-P/Kona95 was distinctly slower compared with 2NS/Kona95. In addition, as time elapsed, high-intensity amide peaks developed for 2NS/Kona95. Aside from a lower-intensity amide peak, 2NS-P/Kona95 also developed a peak at 1635 $cm^{-1}$ recognized as $NH_3^+$ deformation. Physically, 2NS/Kona95 turned into faint yellow, while 2NS-P/Kona95 had no visible change in color towards the end of the exposure. For oxygen-exposed adsorbents, the change in color from white to yellow was an indication that a significant amount of amide was formed [30]. It was revealed that the oxidative degradation of amine occurs via the oxidation of the next methylene groups of the amine to C=O, converting the amine species into amide species [29,35]. To date, there are no further explanations as to why amine groups with propyl spacers are less susceptible to degradations than those with ethyl spacers. In comparison with $CO_2$ degradation, oxidative degradation was expected to give a lower percent decrease in capacity due to the fact that a greater amount of $CO_2$ was fed into the system (approximately 17% $CO_2$ and 3.5% $O_2$ entered the system). To put it briefly, 2NS-P/Kona95, a diamine-containing propyl spacer, is more stable than its counterpart with an ethyl spacer for $O_2$ degradation.

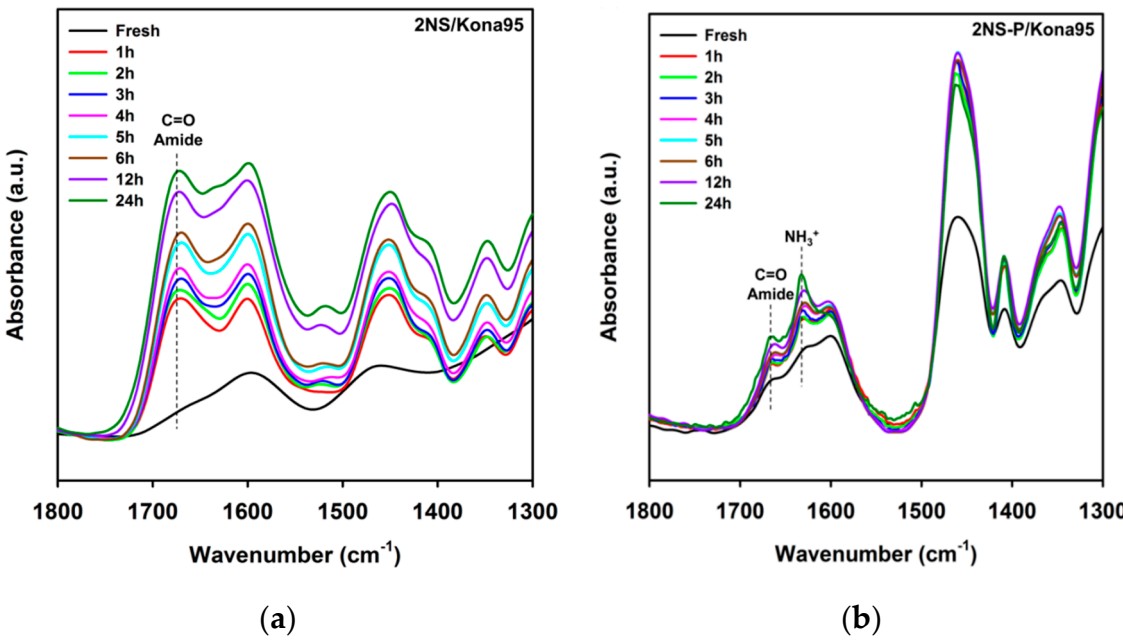

**Figure 6.** Changes in the in situ FT-IR spectra of (**a**) 2NS/Kona95 and (**b**) 2NS-P/Kona95 exposed to 21% $O_2$/balance $N_2$ at 110 °C.

The observation of the formation of degradation species with time was also determined for acidic gases. The changes in the spectrum of the adsorbents per hour are presented in Figure 7 for 50 ppm $SO_2$ and Figure 8 for 50 ppm $NO_2$. Similar to the results for $CO_2$ and oxidative degradations, the spectrum during the first hour changed drastically for both of the adsorbents. More specifically, the peaks attributed to the degradation products appeared already. In fact, the color of 2NS/Kona95 was changed to faint yellow, an indication of amide formation. Then, as the $SO_2$ treatment continued, the peaks grew in intensity and the color of the adsorbent became a darker brown for 2NS/Kona95 and light yellow for 2NS-P/Kona95. Regarding the formation of the nitro group, it is known that the oxidation of nitrogen is preferential to secondary amines [36]. It could be inferred that 2NS-P experienced lower degradation because its longer chain hindered the accessibility to oxidize the secondary amine in its molecule compared with 2NS. Figure 7 shows that new peaks appeared after 24 h of treatment. For 2NS/Kona95, three distinct peaks were developed at 1680, 1650, and 1610 $cm^{-1}$. These peaks were assigned to amide, an NH–$SO_2$ complex, and the nitro group. In comparison, 2NS-P/Kona95 developed only two peaks with lower intensities at 1670 (amide) and 1480 (nitro group) $cm^{-1}$.

The 50 ppm $NO_2$ treatment at 110 °C for 24 h gave results that were similar to the $SO_2$ treatment. A drastic change was evident for the first hour of exposure of both adsorbents to 50 ppm $NO_2$ gas. Then, the peaks gradually increased with time. 2NS/Kona95 was degraded by 40% and 80% when treated with 50 and 200 ppm $NO_2$, respectively (Figure 4b). Interestingly, it appeared that diaminosilane with propyl spacers (2NS-P/Kona95) was also more stable in $NO_2$ degradation, showing only an 11% and a 66% decrease in capacity upon exposure to 50 and 200 ppm of $NO_2$, respectively. Furthermore, the FT-IR spectra of the adsorbents (Figure 8) exposed for 24 h to acidic gas revealed that the decrease in capacity was due to the formation of amide (1670 $cm^{-1}$), an NH–$NO_2$ complex (1650 $cm^{-1}$), and the nitro group (1610 and 1530 $cm^{-1}$). As previously stated, the preferential oxidation of secondary amines by acidic gases to form the nitro group paved way for the increased degradation of 2NS/Kona95 because its shorter chain could provide a more accessible contact for $NO_2$ and secondary amine. Moreover, the increase in $NO_2$ concentration intensified the peaks of the degradation species, suggesting that more degradation products were formed.

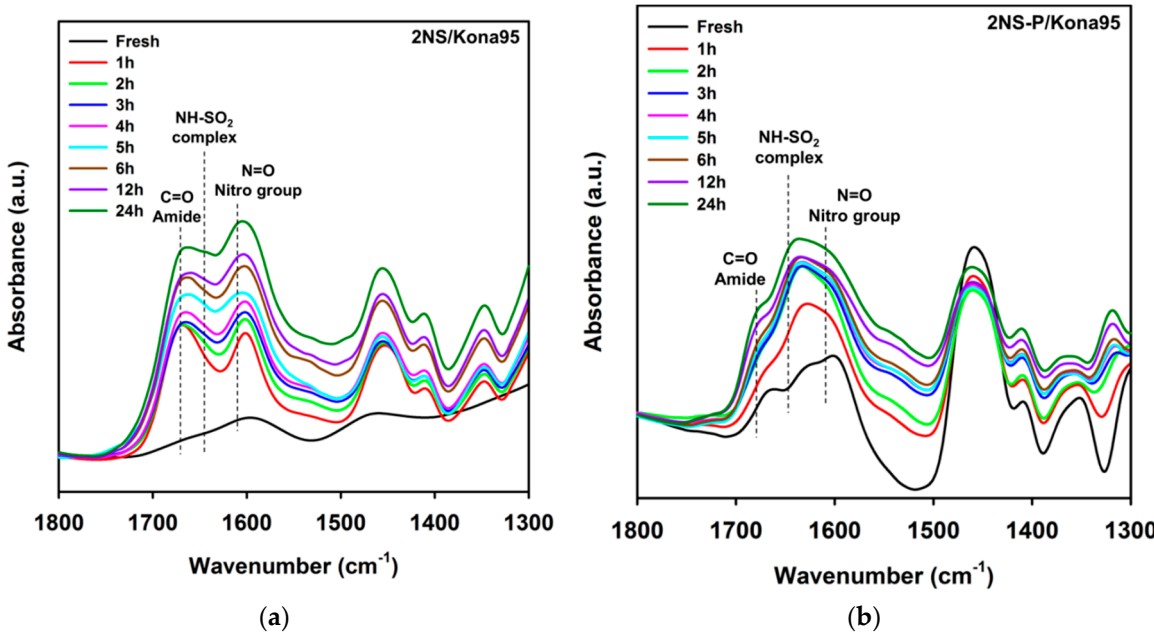

**Figure 7.** Changes in the in situ FT-IR spectra of (**a**) 2NS/Kona95 and (**b**) 2NS-P/Kona95 exposed to 50 ppm $SO_2$ at 110 °C.

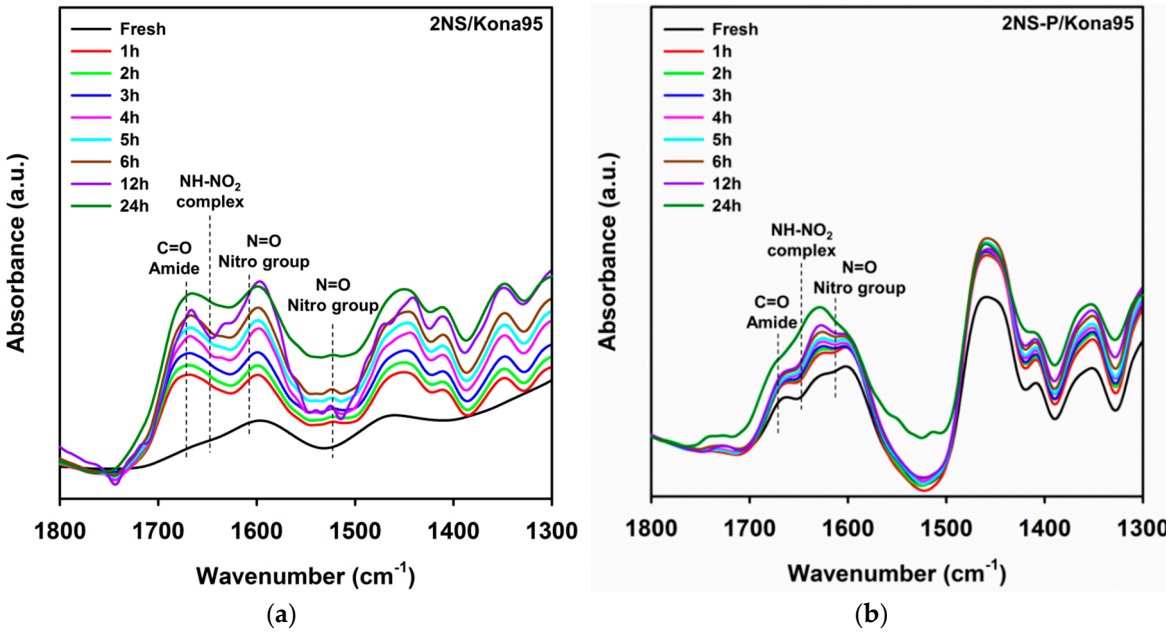

**Figure 8.** Changes in the in situ FT-IR spectra of (**a**) 2NS/Kona95 and (**b**) 2NS-P/Kona95 exposed to 50 ppm $NO_2$ at 110 °C.

At 50 ppm $NO_2$ exposure, similar to $SO_2$ exposure, the degradation species of 2NS/Kona95 were already formed during the first hour (Figure 8a). However, unlike $SO_2$ exposure, the rates of formation of the degradation species for both adsorbents were slower, as evidenced by the minimal increase in peak intensities over time. Furthermore, when the exposure was halted, more pronounced degradation peaks and a higher percent loss in $CO_2$ capacity were observed when the adsorbents were treated with $SO_2$. This was because $SO_2$ is known to be more acidic than $NO_2$ and is more attracted to the basic amine species, thereby yielding deactivation species [37].

The proposed formation reactions of the degradation species are displayed in Figure 9. The formation of the degradation species lowered the $CO_2$ sorption capacities of the adsorbents after hourly exposure of the adsorbents. It was also confirmed that even 50 ppm of $SO_2$ could affect the stability of amine-functionalized adsorbents.

**Figure 9.** Proposed mechanisms for the formation of (**a**) amide with $O_2$, (**b**) NH–$NO_2$ complex, (**c**) nitro groups and amide with acidic gases, and (**d**) NH–$SO_2$ complex with acidic gases.

Among all the induced degradation gases, $CO_2$ caused the highest percentage loss in capacity, even after regeneration, due to its relatively higher concentration in the gas stream. Even so, the presence of more acidic gases ($SO_2$ and $NO_2$) in minute amounts could cause a relatively high catalytic deactivation of the amine-functionalized adsorbents. In comparison, diamine with propyl spacers was more stable than its counterpart with ethyl spacers, possibly due to the steric hindrance caused by the longer chain. Additional –$CH_2$ might either block the active centers of the molecule or could impose stronger forces, so that the typical reaction pathways could not proceed.

## 3. Materials and Methods

The silica support, Kona95, was used in the synthesis of the adsorbents. Kona95 utilized fumed silica (KONASIL K-300, OCI, Gunsan, Korea) and silica sol (Ludox AS-30, Aldrich, Saint Louis, USA). The aminosilane with ethyl spacers was N-[3-(trimethoxysilyl)propyl]ethylenediamine (2NS, 97% purity, Merck, Saint Louis, USA). The diaminosilane with propyl spacers was prepared by reacting (3-chloropropyl)trimethoxysilane (CPTMS, 97% purity, Alfa Aesar,

Ward Hill, MA, USA) and 1,3-diaminopropane (DAP, 98% purity, Alfa Aesar) to produce $N^1$-(3-(trimethoxysilyl)propyl)propane-1,3-diamine (2NS-P). The structures of these materials are shown in Table 4. KBr (PIKE Technologies, Fitchburg, WI, USA) powder was stored in the glovebox. Nitrogen gas (99.999%, Daedeok Gas, Incheon, Korea), $CO_2$ (Daedeok Gas), Air (21% $O_2/N_2$, Daedeok Gas), $SO_2$ (50 ppm, Daedeok Gas & 200 ppm, Union Gas Seoul, Korea), and $NO_2$ (50 ppm, Daedeok Gas & 200 ppm, Union Gas) were all ultra-high-purity grade. The nitrogen gas used during the preparation of the adsorbents was further purified using a Fisher RIDOX column and a molecular sieve 5A/13X (Aldrich) column.

**Table 4.** Structures of 2NS and the precursors of 2NS-P.

| Material | Structure |
|---|---|
| N-[3-(trimethoxysilyl)propyl]ethylenediamine (2NS) |  |
| (3-Chloropropyl)trimethoxysilane (CPTMS) |  |
| 1,3-Diaminopropane (DAP) |  |
| $N^1$-(3-(trimethoxysilyl)propyl)propane-1,3-diamine (2NS-P) |  |

The silica support, Kona95, was synthesized via spray-drying of a slurry prepared by mixing 0.95 kg of fumed silica, 0.05 kg of silica sol, and 9 kg of water. The resulting silica was calcined in dry air at 600 °C and was stored in the glovebox.

For the adsorbent preparation, a 250 mL round-bottom flask containing 1.0 g of Kona95 was attached to a Schlenk line to perform the procedures under nitrogen atmosphere. The aminosilane (2NS or 2NS-P; 6 mmol) was added dropwise to the Kona95 while stirring gently. Then, the temperature was raised to 50 °C for the reaction to proceed for 3 h. The functionalized Kona95 preparations are denoted as 2NS/Kona95 and 2NS-P/Kona95.

The thermal stability and $CO_2$ adsorption performance determinations were carried out using a thermogravimetric analyzer (TGA, SDT Q600, TA Instruments, New Castle, DE, USA). The $CO_2$ sorption capacities before and after the exposure to acidic gases were determined by exposing the adsorbents to 60 mL/min of $N_2$ at 110 °C for 1 h as a pretreatment procedure. Then, the gas flow was changed to a 17% $CO_2/N_2$ mixture gas for 1 h at 30 °C to determine the $CO_2$ sorption capacities of the adsorbents.

For the FT-IR analyses, samples were formed into discs by applying 3 t of pressure to the sample in a pellet die that was 13 mm in diameter. Prior to IR spectra collection, the samples were degassed at 110 °C under a vacuum condition for 1 h. Then, the system was cooled to room temperature and the IR spectra were recorded with 32 scans at a 4 cm$^{-1}$ resolution. During the 24 h of exposure of the adsorbents to gases, the gas flow was halted hourly and the system was vacuumed for 5 min to remove physically adsorbed species; then, the FT-IR spectrum was measured. The exposure gases were 100% $CO_2$, 21% $O_2$/balance $N_2$, $SO_2$ (50 and 200 ppm), and $NO_2$ (50 and 200 ppm).

## 4. Conclusions

Silica was modified with diaminosilanes with ethyl and propyl spacers via the incipient wetness technique. The resulting adsorbents were evaluated via TGA and FT-IR spectroscopy to determine the extent of degradation at 110 °C for 24 h with $CO_2$, $O_2$, $SO_2$, and $NO_2$.

$CO_2$ was found to form more cyclic urea with the 2NS-functionalized catalytic silica than the 2NS-P-functionalized one. Moreover, degradations induced by $O_2$, $SO_2$, and $NO_2$ were worse in 2NS-functionalized than in 2NS-P-functionalized catalytic silica. It was proved that amide was not just a degradation product from $O_2$ but also from acidic gases. Additionally, acidic gases could also oxidize nitrogen atoms of amines to form a nitro group and they were capable of forming complexes with amines. These deactivation species were formed even during the first hour of exposure, so it is important to note that by shortening the cycling time at this temperature, the formation of these species could be minimized.

These outcomes suggest that the conventional molecules used in the preparation of adsorbents for $CO_2$ capture may be replaced by those which have a comparable number of nitrogen atoms, with a longer spacing in between the nitrogen atoms.

**Author Contributions:** Conceptualization, R.M.P. and Y.S.K.; methodology, R.M.P. and Y.S.K.; writing—original draft preparation, R.M.P.; writing—review and editing, C.M. and Y.S.K.; supervision, Y.S.K.; project administration, Y.S.K.; funding acquisition, Y.S.K.

**Funding:** This work was supported by a National Research Foundation of Korea (NRF) grant funded by the Korean government (MSIP) (2016R1D1A1B01009941). This work was also supported by the Human Resources Program in Energy Technology of the Korea Institute of Energy Technology Evaluation and Planning (KETEP), granted financial resources from the Ministry of Trade, Industry & Energy, Republic of Korea (No. 20194010201730).

**Conflicts of Interest:** The authors declare no conflict of interest.

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
