# Peer review of "In Situ IR Study on Effect of Alkyl Chain Length between Amines on Its Stability against Acidic Gases"

_catalysts, doi:10.3390/catal9110910_

Round 1

Reviewer 1 Report

The presentation of the article can be improved. For example, sentences like "Numerous researches have been made regarding the effect of increasing the number amine groups but only a few have reported the difference of amine groups with propyl spacer compared to ethyl spacer" is not very clear and I had to read it several times to guess what the authors wanted to say.

"According to Pang et al., PPI based CO2 adsorbents showed increased tolerance during regeneration step and the presence of secondary amines does not necessarily yield to degradation species at elevated temperatures" what does PPI stand for?

"It is quite important to study on and improve the stability of amine adsorbents against acidic gases and CO2 for both industrial and academic purposes." It should be specified more clearly that these adsorbents are amine-based and not amine, which is confusing for someone not dealing with this very specific topic.

Experimental:

The motives for selecting the procedures described are not clear to me from the manuscript. For example: What is the practicality of measuring the adsorption capacity for CO2 after the exposition of the adsorbent to 100% CO2 for several hours at elevated temperature? Sentences like "Much attention has been drawn to the effect of CO2 to adsorbents primarily because huge percentage of the flue gas is composed of CO2" do not clarify the situation at all (on the contrary) – I guess the authors want to capture this CO2 from the flue gas? The concentration in the flue gas is normally between 10%–20%, but you would not want to capture the CO2 from the gas containing only CO2? On the contrary, during the release stage (of the CO2 captured), the concentration of CO2 should be very high (almost 100%), do the authors want to simulate this?

Also from my viewpoint, what matters most, in the case of CO2 adsorbents, is their cyclic performance. I do not see much sense in capturing CO2, then let the adsorbent be under the atmosphere of 100% CO2 at elevated temperature, and then capture the CO2 again.

Reviewer 2 Report

In this article, Ko and coworkers describe an in-situ IR study of amine-functionalized silica for CO2 capture and study the effect of alkyl chain length on their stability under various conditions. The material is of general interest, but the pertaining process, CO2 adsorption/desorption, is not catalytic, thus in my view, it is not suitable for publication in Catalysts.   

Here are a few points that could be considered to improve the manuscript:

2NS/Kona and 2NS-P/Kona should be better designated more explicitly and earlier in the manuscript.

It is surprising that the CO2 sorption correlates proportionally with the surface area, but inverse proportionally with the silane content. One would think that higher silane content, which carries the amine unit, should lead to higher sorption of CO2.

Equations in Figure 8 may need some explanation. For example, how does NH-NO2 complex form in b, where does the positive charge come from?  

Table 1 caption is the same as Table 2 and should be corrected.

Reviewer 3 Report

This study reported the amine-functionalized silica for CO2 capture of diamines with ethyl and propyl spacer and the degradation species formed after long-term exposure to various acidic gases. The results provided scientific and methodological soundness. But there are some issues should be addressed before its acceptance.

Please provide more novelty statements in the Introduction about the silica support, and why functionalization was presented in this study.  The authors should provide more previous papers to demonstrate the background. Why the BET surface area decreased significantly after amine incorporation? Please supplement the data of Kona95 in Table 2 and all the Figures. I found there is no Figure 3, and two Figure 8. Please provide more solid evidence to prove the presence of amide, NH-NO2 complex, and NH-SO2 complex. NMR should be measured. I suggest discussing the sample from the viewpoint of catalysis to meet the requirement of the JOURNAL.

Round 2

Reviewer 1 Report

Almost all of my previous concerns were addressed. Only some explanation if the selected methodology will be transferable to a real case, where a cyclic performance will be carried out is missing in my opinion.

Reviewer 2 Report

The paper is acceptable. 

Reviewer 3 Report

The paper has been revised carefully. 

I suggest accept this manuscript in the present form without further reviewing.

This manuscript is a resubmission of an earlier submission. The following is a list of the peer review reports and author responses from that submission.